# Effect of Different Carbohydrate Intakes within 24 Hours after Glycogen Depletion on Muscle Glycogen Recovery in Japanese Endurance Athletes

**DOI:** 10.3390/nu14071320

**Published:** 2022-03-22

**Authors:** Keiko Namma-Motonaga, Emi Kondo, Takuya Osawa, Keisuke Shiose, Akiko Kamei, Motoko Taguchi, Hideyuki Takahashi

**Affiliations:** 1Japan Institute of Sports Sciences, Tokyo 115-0056, Japan; akiko.kamei@jpnsport.go.jp (A.K.); takahashi.hideyuk.ga@u.tsukuba.ac.jp (H.T.); 2Japan Society for the Promotion of Science, Tokyo 102-0083, Japan; kondo.emi.fp@u.tsukuba.ac.jp; 3Faculty of Health and Sport Sciences, University of Tsukuba, Tsukuba 305-8574, Japan; 4Faculty of Physical Education, Japan Women’s College of Physical Education, Tokyo 157-8565, Japan; takuya.osawa.jp@gmail.com; 5Faculty of Education, University of Miyazaki, Miyazaki 889-2192, Japan; kshiose@cc.miyazaki-u.ac.jp; 6Faculty of Sport Sciences, Waseda University, Saitama 359-1192, Japan; mtaguchi@waseda.jp; 7Advanced Research Initiative for Human High Performance (ARIHHP), University of Tsukuba, Tsukuba 305-8574, Japan

**Keywords:** glycogen, recovery, carbohydrate, ^13^C-magnetic resonance spectroscopy, exercise, muscle

## Abstract

Daily muscle glycogen recovery after training is important for athletes. Few studies have reported a continuous change in muscle glycogen for 24 h. We aimed to investigate the changes in carbohydrate intake amount on muscle glycogen recovery for 24 h after exercise using ^13^C-magnetic resonance spectroscopy (^13^C-MRS). In this randomized crossover study, eight male participants underwent prolonged high-intensity exercise, and then consumed one of the three carbohydrate meals (5 g/kg body mass (BM)/d, 7 g/kg BM/d, or 10 g/kg BM/d). Glycogen content of thigh muscle was measured using ^13^C-MRS before, immediately after, and 4 h, 12 h and 24 h after exercise. Muscle glycogen concentration decreased to 29.9 ± 15.9% by exercise. Muscle glycogen recovery 4–12 h after exercise for the 5 g/kg group was significantly lower compared to those for 7 g/kg and 10 g/kg groups (*p* < 0.05). Muscle glycogen concentration after 24 h recovered to the pre-exercise levels for 7 g/kg and 10 g/kg groups; however, there was a significant difference for the 5 g/kg group (*p* < 0.05). These results suggest that carbohydrate intake of 5 g/kg BM/d is insufficient for Japanese athletes to recover muscle glycogen stores 24 h after completing a long-term high-intensity exercise.

## 1. Introduction

Muscle glycogen is an important source of energy during exercise [1] and affects endurance performance [2]. Exercise-induced muscle glycogen consumption is restored by ingesting sufficient amounts of carbohydrates [3,4,5], and their importance is indicated in the sports nutrition position statement [6] and International Association of Athletics Federations Consensus Statement [7].

Muscle glycogen recovery over a 24 h period is also important for athletes to maintain their daily training quality. According to the international guidelines [6], the recommended amount of carbohydrate intake per day is 3–5 g/kg body mass (BM)/d for low-intensity or skill-based training programs, 5–7 g/kg BM/d for moderate-intensity training programs, 6–10 g/kg BM/d for endurance training programs, and 8–12 g/kg BM/d for extreme training programs.

Previous studies have reported a relationship between carbohydrate-rich meals (including beverages) and 24 h muscle glycogen recovery [8,9,10,11,12,13,14,15,16]. There was no difference in muscle glycogen recovery with the consumption of different meals that have the same carbohydrate content [11]. A high glycemic index (GI) diet restores muscle glycogen faster [9]. Moreover, high-carbohydrate diets recover muscle glycogen more efficiently than high-fat diets [13]. With sufficient carbohydrate intake, fat and protein intake does not affect muscle glycogen recovery [10]. However, most of these studies investigated this relationship in a Caucasian population. Ethnic differences, body size, shape, and food culture affect the metabolic responses to carbohydrates [17,18]. There have been studies comparing the racial physique of Caucasian American and Japanese university students [19], as well as that of Australian Caucasian and Japanese young adults [20]. Both studies reported that Japanese people weigh less. A national nutrition survey reported that the mean carbohydrate intake of American women and American men aged 20–29 years is 230 g and 271 g, respectively [21]. In contrast, the mean carbohydrate intake of Japanese women and Japanese men of the same age is 213 g and 293 g, respectively [22]. In this survey, the mean BM was more in the American population (women, 74.9 kg; men, 85.5 kg) [23] than in the Japanese population (women, 53.0 kg; men, 65.2 kg). Therefore, the relative daily carbohydrate intake per kg BM is presumed to be higher in the Japanese population than in the American population. However, it is unclear whether the international guidelines for muscle glycogen recovery can be adapted to Japanese athletes.

Additionally, previous studies used the muscle biopsy procedure for analyzing muscle glycogen content 24 h after exercise. The invasive nature of a muscle biopsy and repeated measurements can disturb the muscle glycogen content. Inflammation after muscle biopsy can further impair glycogen synthesis for a few days [8]. Hence, time-course changes of muscle glycogen recovery during a 24 h period are not well understood.

^13^C-magnetic resonance spectroscopy (^13^C-MRS) was established as an alternative muscle glycogen measurement system [24,25,26,27]. It is non-invasive and facilitates repeated muscle glycogen measurements in the same individual.

The purpose of this study was to investigate the effect of three different amounts of carbohydrate intake on muscle glycogen recovery rates using ^13^C-MRS.

## 2. Materials and Methods

### 2.1. Participants

Eight male collegiate endurance athletes participated in this study. They were recruited from four Japanese college teams. Their mean ± standard deviation age, height, BM, body fat percentage, and maximal oxygen uptake ($\dot{V}$O_2_max) were 20 ± 1 year, 167.5 ± 6.5 cm, 56.2 ± 6.9 kg, 9.4 ± 3.7%, and 55.8 ± 6.0 mL/kg/min, respectively. The study protocol was approved by the Ethics Committee of Japan Institute of Sports Sciences (approval number: JISS-IRB-2013-008). All participants provided written informed consent before the collection of the data.

### 2.2. Experimental Design

This study was conducted as a randomized crossover trial. The participants reported to the laboratory before the first trial to complete a maximal incremental exercise test until they were exhausted, and their $\dot{V}$O_2_max was measured using a cycle ergometer (Aerobike 75XLII; Combi, Tokyo, Japan). The test consisted of a 3 min warm-up at 40 W, followed by a program that increased by 20 W every minute, until voluntary exhaustion. Oxygen consumption was measured using an indirect calorimeter, and $\dot{V}$O_2_max was calculated according to a previous study [27]. For the assessment of habitual nutrient intake of energy and macronutrients (carbohydrate, protein and fat) and habitual energy expenditure, three-day dietary and one-day physical activity records were conducted before the experimental trial.

The eight participants visited the laboratory for three experimental trials separated by at least seven days in a counterbalanced, randomized crossover design. On the experimental days, blood samples of the participants were collected, and their muscle glycogen concentration was measured before exercise. Thereafter, participants completed the glycogen depletion exercise of approximately 90 min [27]. Subsequently, they consumed one of the three carbohydrate meals containing different amounts of carbohydrate (5 g/kg BM/d (5 g), 7 g/kg BM/d (7 g), or 10 g/kg BM/d (10 g)) during the 24 h recovery period after exercise. All participants performed three trials. The order of 5 g, 7 g and 10 g dietary trials varied from participant to participant. Muscle glycogen concentrations were measured using ^13^C-MRS before, immediately after, and 4 h, 12 h and 24 h after exercise. Plasma glucose, serum insulin, and plasma glucagon concentrations were also measured before and immediately after exercise, as well as 30 min after lunch, dinner, and breakfast (Figure 1).

### 2.3. Habitual Intakes of Energy and Macronutrients and Habitual Energy Expenditure

Before the measurements, participants completed weighed dietary records for three separate days, including two training days and one day off. The habitual intakes of energy and macronutrients (carbohydrate, protein and fat) were calculated by the same method as that of previous studies [27,28]. The habitual energy expenditure of the participants was estimated before the experimental trial using a 3-axis accelerometer (Actimarker, Panasonic, Osaka, Japan).

### 2.4. Dietary Intervention

All three experimental meals provided the same energy content. The energy and macronutrients content of the three experimental meals are presented in Table 1. After completing the glycogen depletion exercise, participants consumed the following three meals: lunch (13:00), dinner (19:00), and breakfast (07:00 the next day). Water was available *ad libitum* throughout the study. Muscle glycogen measured 4 h after exercise corresponds to 3 h after lunch. The carbohydrate intakes per hour for the 5 g, 7 g and 10 g meals were 0.6 g/kg BM, 0.9 g/kg BM and 1.2 g/kg BM, respectively.

### 2.5. Muscle Glycogen Concentration

Similar to a previous study [26,27], all measurements of the muscle glycogen concentration were performed on the thigh muscle group of the right leg: the vastus lateralis and vastus intermedius muscles. ^13^C-MRS was performed using a 3 T superconducting MR scanner (Magnetom Verio, Siemens, Erlangen, Germany) with a 70 cm horizontal bore, providing an operating frequency of 123.2 MHz The ^13^C-MRS spectra were collected using a ^13^C-^1^H double-tuned 10 cm-diameter surface coil placed on the vastus lateralis muscle. Measurement parameters for ^13^C MRS were: repetition time = 200 ms, data points = 2048, sweep width = 20,000 Hz. We applied a proton decoupling (WALTZ-4, Duration = 1 ms, Flip angle = 180 deg, decoupling during 50% of the acquisition window) and nuclear overhauser enhancement (Rectangular, Duration = 5 ms, Flip angle = 90 deg) technique. Data were obtained as the sums of 4500 free induction decays (FIDs). The muscle glycogen concentrations were determined by comparison with an external standard solution (150 mM glycogen from oyster + 50 mM KCl) in a cylindrical phantom the size of which was identical to the muscle in each subject. Peak areas were integrated using commercial software provided with the MR console. The coefficient of variation (CV) of repeat measurement of muscle glycogen by ^13^C MRS with repositioning and reshimming was 3.5 ± 2.1% in our preliminary experiment.

### 2.6. Blood Analysis

At pre- and post-exercise, and 30 min after lunch, dinner, and breakfast, venous blood samples were collected from the antecubital vein into vacuum tubes containing dipotassium ethylenediamine tetraacetic acid and serum separator tubes. Serum glucose, insulin, and plasma glucagon concentrations were measured. The venous blood samples were outsourced to an independent laboratory (LSI Medience Corporation, Tokyo, Japan).

### 2.7. Statistical Analysis

All data are presented at means ± standard deviation unless otherwise stated. A repeated measures analysis of variance (ANOVA) with two-within-subjects factors (time and amount of carbohydrates) was used for muscle glycogen concentration and blood parameters to evaluate interaction. When significant interaction effects were observed, the muscle glycogen concentration was subsequently analyzed using Dunnett’s test to evaluate within-group difference to pre-exercise value. The recovery rate of the muscle glycogen concentration and blood parameters were subsequently analyzed using Bonferroni’s test to evaluate between-group differences at each time point. The plasma glucose 5 g—12 h and the serum insulin 10 g—pre are deficient one data. All statistical analyses were performed using IBM SPSS Statistics version 24.0 for Windows (IBM Corp., Armonk, NY, USA).

## 3. Results

### 3.1. Habitual Intakes of Energy and Macronutrients

The habitual intakes of energy, protein, fat, and carbohydrate were 2685 ± 524 kcal, 97 ± 25 g, 81 ± 23 g and 377 ± 89 g (carbohydrate: 6.8 ± 1.7 g/kg BM/d), respectively. The energy rates of protein, fat and carbohydrate were 14 ± 2%, 27 ± 7% and 56 ± 6%, respectively.

### 3.2. Muscle Glycogen Concentration

There was a significant interaction between the effects of carbohydrate contents and time for muscle glycogen synthesis (F (8, 56) = 2.550, *p* = 0.019). The muscle glycogen concentration before exercise was similar for all the three carbohydrate intake amounts, and similarly decreased by the glycogen depletion exercise to 27.3 ± 9.9%, 29.4 ± 18.0% and 33.1 ± 19.6% of the pre-exercise value in the 5 g, 7 g and 10 g groups, respectively (Figure 2). The muscle glycogen concentration recovered to 81.7 ± 21.8% (recovery volume: 36.5 ± 14.4 mmol/kg BM wet weight), 97.1 ± 16.1% (45.6 ± 15.6 mmol/kg BM wet weight) and 100.1 ± 12.9% (46.2 ± 14.9 mmol/kg BM wet weight) of the pre-exercise levels at 24 h after exercise for the 5 g, 7 g and 10 g groups, respectively. After 24 h, the muscle glycogen concentration recovered to the pre-exercise levels in the 7 g and 10 g groups, although there was a significant difference for the 5 g group (*p* < 0.05).

Figure 3 shows the amount of muscle glycogen recovered for a specific period (i.e., recovery rate). There was a significant interaction between the time and amount of carbohydrates (F (4, 28) = 4.087, *p* = 0.010). For the 4–12 h period, the muscle glycogen recovery after exercise was significantly lower in the 5 g than in the 7 g and 10 g groups (*p* < 0.05).

### 3.3. Blood Analysis

The changes in plasma glucose, serum insulin, and plasma glucagon concentrations are presented in Figure 4, respectively. A significant interaction between the time and amount of carbohydrates was found for serum insulin (F (8, 48) = 5.208, *p* < 0.001) and plasma glucagon concentration (F (8, 56) = 4.706, *p* < 0.001), but not for the plasma glucose concentration. No significant difference was observed in the plasma glucose concentration for the three dietary intervention groups. Serum insulin concentrations were similar before and after exercise in all three groups. However, there was a significant difference in the serum insulin concentration between the 5 g and 10 g groups after lunch (*p* < 0.05). After dinner, the serum insulin concentration of the 7 g group was also higher than that of the 5 g group (*p* < 0.05). Plasma glucagon concentrations of the three groups were similar until after dinner. After breakfast, the plasma glucagon concentration was significantly higher in the 7 g group than in the 10 g group (*p* < 0.05). Furthermore, the plasma glucagon concentration of the 5 g group was higher than that of the 10 g group, although this difference was not significant (*p* = 0.076).

## 4. Discussion

This study investigated the differences in muscle glycogen recovery in Japanese male endurance athletes 24 h after completion of a glycogen depletion exercise. Post-exercise, they consumed three meals containing different amounts of carbohydrates (5 g, 7 g and 10 g/kg BM/d). Our study is novel as the time-course changes in muscle glycogen concentration were measured repeatedly over 24 h using ^13^C-MRS instead of an invasive procedure such as muscle biopsy; therefore, inter-individual differences were eliminated.

Our results indicated that the amount of muscle glycogen 24 h after glycogen depletion exercise was not significantly different from the pre-exercise values for the 7 g and 10 g groups, but remained significantly lower for the 5 g group (Figure 2). This is consistent with a position statement about sports nutrition [6]. Kiens and Richter [14] reported that muscle glycogen decreased to 20.2% after exercise and recovered by 76.7% after 18 h and 93.8% after 30 h on consuming 8.3 g/kg BM of carbohydrates. Starling et al. [13] reported a muscle glycogen recovery of 54.6% at 1.9 g/kg BM and 96.1% at 9.8 g/kg BM, 24 h after exercise. The results of the present study corroborate with these previous studies, which assessed 24 h recovery after measuring muscle glycogen before exercise.

The time-course changes in muscle glycogen concentration were measured over a 24 h period. In addition, the muscle glycogen recovery for each time segment was compared between 5 g and the other carbohydrate intakes to examine the difference in muscle glycogen recovery. As shown in Figure 3, there was no difference in muscle glycogen recovery between the three dietary intake groups during the first 4 h. This observation is similar to previous a study by Ivy et al. [29], suggesting that the amount of a glucose supplement does not affect glycogen restoration 2 h and 4 h post-exercise. After glycogen is depleted, resynthesis is mostly affected by glucose uptake induced by muscle contraction, rather than the amount of carbohydrate consumed [24]. On the other hand, the recovery of the 5 g group was significantly lower 4–12 h after exercise than that of the 7 g and 10 g groups. The lower muscle glycogen recovery rate for the 5g group 24 h after glycogen depletion exercise was possibly due to the low recovery observed 4–12 h after exercise.

The difference in the muscle glycogen concentration 24 h after exercise suggests that the postprandial plasma glucose and serum insulin concentrations influence the muscle glycogen synthesis ratio [16,30]. There was no significant difference in the plasma glucose concentration among the three dietary intake groups (Figure 4A), and the serum insulin concentrations of the 10 g group after lunch and 7g group after dinner were higher than that of the 5 g group (Figure 4B). Thus, increasing the blood glucose and serum insulin concentrations accelerated insulin-dependent glucose uptake 4–12 h after exercise [30]. No significant difference in muscle glycogen recovery between the 7 g and 10 g groups was observed, which may be due to the insulin-dependent glucose uptake after dinner. Hence, it can be considered that insulin-dependent glucose uptake of the 7 g group was accelerated 4–12 h after exercise, resulting in the restoration of muscle glycogen to the same concentration as that of the 10 g group. Moreover, the plasma glucagon concentration of the 5 g group 24 h post-exercise tended to be higher than that of the 10 g group (Figure 4C). Glucagon is secreted from the pancreas, which increases blood glucose levels by inhibiting glycogenesis in the liver. The increase in blood glucagon concentration of the 5 g group 24 h post-exercise could be due to compensation for the lack of blood glucose in the body. This supports the conclusion that a 5 g/kg BM intake of carbohydrates is insufficient after a glycogen depletion exercise.

In this study, participants recover their muscle glycogen content by consuming typical Japanese meals, rather than sports foods. Costill et al. [31] reported that the carbohydrate sources for their study were sugar, starch, and drinks, whereas Burke et al. [10] used cornflakes, whole wheat bread, instant mashed potato, and Polycose^®^ as test dishes. On the other hand, rice, pasta, bread, and fruits were used as carbohydrate sources in this study. The participants were provided with a Japanese-style meal, which is a combination of staple food that contains a carbohydrate source, a main dish consisting of meat and fish as a protein source, and a side dish that contains vegetables. As a result, many meals with a low GI may have been offered.

Western diets are said to be difficult to ingest due to the large volume of normal diets to ensure a high carbohydrate content [11]. The effects of a high GI diet are similar to that of Western dietary patterns [30], which may differ from the Asian diet that is abundant in rice. Considering the dietary habits of athletes, it is necessary to consider their muscle glycogen recovery, as they are unlikely to obtain a sufficient intake of other nutrients with each meal, with drinks or snacks alone.

Ethnic differences may affect muscle glycogen recovery [17,18]. Some previous studies have shown that there are differences in the metabolic response to carbohydrate intake for different ethnicities [32,33]. Furthermore, the amount of carbohydrates athletes eat daily may also have an effect [34]. It is difficult to compare our findings about muscle glycogen recovery with previous findings, as only a few studies have reported the habitual carbohydrate intake before the trial. However, reports of higher carbohydrate intake in the Japanese population compared to that of the American population indicate that future strategies investigating muscle glycogen storage in the Asian population need to be considered [21,22].

This study had some limitations. First, meal regulation was a limitation. Increasing the intake of carbohydrates increases the amount of total energy intake [15]. For expressing the amount of energy intake as habitual energy intake, the energy intake, which is influenced by the carbohydrate content was adjusted as the fat content. As high-fat meals affect digestion and absorption, they are considered to be inappropriate for athletes [34]. Thus, digestion and absorption may influence muscle glycogen recovery. Additionally, in short-term muscle glycogen recovery, an adequate dietary carbohydrate intake has been shown to increase muscle glycogen synthesis [35]. In this study, participants consumed approximately the same amount of protein in all three diets (2.0 g/kg BM/d), which is in agreement with the position statement [6].

## 5. Conclusions

Our results suggest that a carbohydrate intake of 5 g/kg BM/d is insufficient for muscle glycogen recovery 24 h after completing a glycogen depletion exercise in Japanese endurance athletes who habitually consume a large amount of carbohydrates. In addition, the results of the ^13^C-MRS revealed lower glycogen synthesis in the 5 g group, which occurred 4–12 h after exercise, and it may relate to lower plasma glucose and serum insulin concentrations after a meal.

## Figures and Tables

**Figure 1 nutrients-14-01320-f001:**
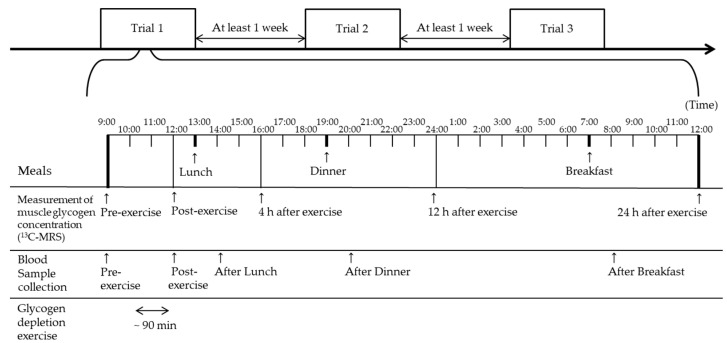
Experimental design of the study.

**Figure 2 nutrients-14-01320-f002:**
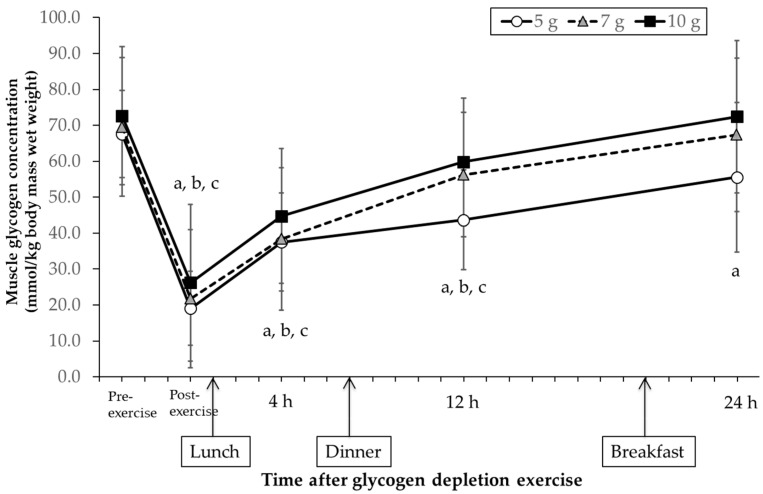
Changes in muscle glycogen concentration over 24 h for the three carbohydrate intake amounts—a: 5 g group, b: 7 g group, c: 10 g group, vs. pre-exercise; *p* < 0.05; values are expressed as mean ± standard deviation.

**Figure 3 nutrients-14-01320-f003:**
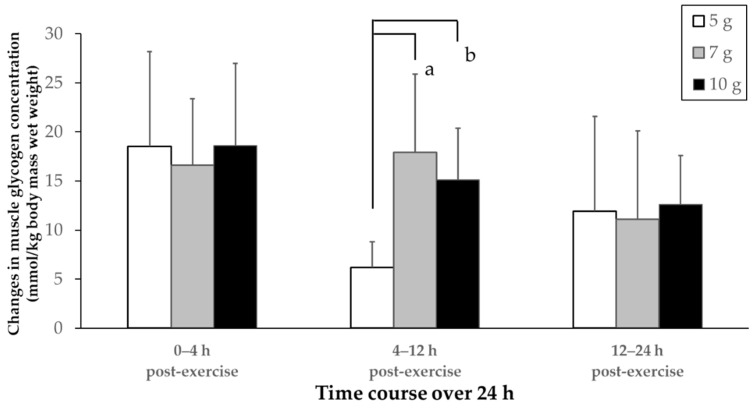
Recovery rate of muscle glycogen concentration over 24 h for the three carbohydrate intake amounts—a: 5 g vs. 7 g groups, *p* < 0.05; b: 5 g vs. 10 g groups, *p* < 0.05; values are expressed as mean ± standard deviation.

**Figure 4 nutrients-14-01320-f004:**
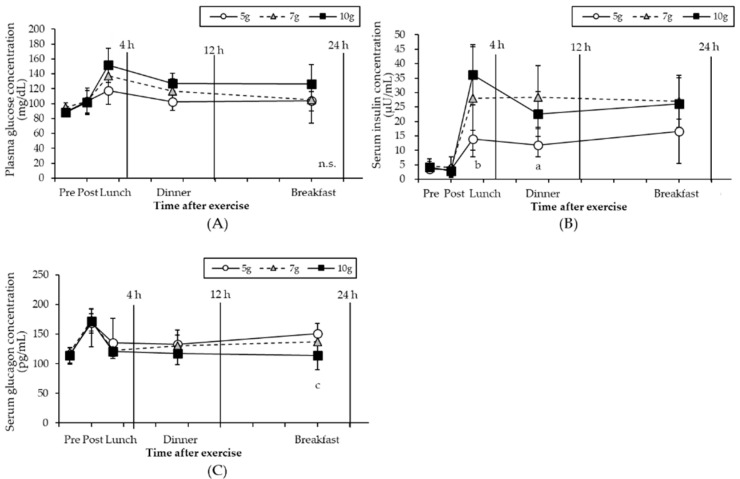
Changes in plasma glucose (**A**), serum insulin, (**B**), and plasma glucagon levels (**C**) Values are expressed as mean ± standard deviation. a: 5 g vs. 7 g groups, *p* < 0.05; b: 5 g vs. 10 g groups; *p* < 0.05; c: 7 g vs. 10 g groups, *p* < 0.05, n.s.: not significant.

**Table 1 nutrients-14-01320-t001:** Energy and macronutrients (carbohydrate, fat and protein) contents of the experimental meals. Values are expressed as means ± standard deviation.

Meal		5 g Meal	7 g Meal	10 g Meal
Energy (kcal)	Lunch	1159 ± 102	1162 ± 117	1169 ± 111
	Dinner	1159 ± 100	1158 ± 106	1170 ± 120
	Breakfast	868 ± 88	861 ± 79	868 ± 71
	Total	3186 ± 286	3181 ± 297	3207 ± 297
Carbohydrate (g)	Lunch	106 ± 16	146 ± 17	205 ± 25
	Dinner	102 ± 11	142 ± 20	201 ± 30
	Breakfast	77 ± 10	108 ± 13	157 ± 14
	Total	285 ± 37	396 ± 49	564 ± 67
Carbohydrate (g/kg)	Lunch	1.9 ± 0.1	2.6 ± 0.1	3.7 ± 0.1
	Dinner	1.8 ± 0.1	2.5 ± 0.1	3.6 ± 0.2
	Breakfast	1.4 ± 0.1	1.9 ± 0.1	2.8 ± 0.1
	Total	5.1 ± 0.2	7.1 ± 0.2	10.0 ± 0.2
Protein (g)	Lunch	42 ± 5	41 ± 5	36 ± 7
	Dinner	42 ± 5	43 ± 4	39 ± 5
	Breakfast	29 ± 3	30 ± 3	25 ± 4
	Total	113 ± 12	114 ± 12	100 ± 15
Protein (g/ kg)	Lunch	0.7 ± 0.0	0.7 ± 0.0	0.7 ± 0.1
	Dinner	0.7 ± 0.0	0.8 ± 0.0	0.7 ± 0.1
	Breakfast	0.5 ± 0.0	0.5 ± 0.0	0.5 ± 0.1
	Total	2.0 ± 0.0	2.0 ± 0.1	1.8 ± 0.4
Fat (g)	Lunch	61 ± 17	45 ± 19	22 ± 18
	Dinner	64 ± 16	44 ± 19	22 ± 21
	Breakfast	49 ± 15	35 ± 14	15 ± 12
	Total	174 ± 47	125 ± 52	58 ± 50

## Data Availability

Not applicable.

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
