# Peer review of "Effect of Different Carbohydrate Intakes within 24 Hours after Glycogen Depletion on Muscle Glycogen Recovery in Japanese Endurance Athletes"

_nutrients, 2022, doi:10.3390/nu14071320_

Round 1

Reviewer 1 Report

Review: Effect of different carbohydrate intakes within 24 hours after glycogen depletion on muscle glycogen recovery in Japanese endurance athletes

The authors tracked the glycogen content of thigh skeletal muscle in 8 male college endurance athletes using 13C-magnetic resonance spectroscopy. Glycogen was measured before, immediately after and at 3 time points (up to 24 hours) after prolonged high intensity exercise, with the experimental variable being the amount of carbohydrates consumed in three post-exercise meals (5g/kg/d, 7g/kg/d and 10g/kg/d carbohydrates).

Concerns

My biggest concern is that it states that a randomized crossover trial was used which makes me think that all 8 athletes performed the experiment 3 times, each time being given a different carbohydrate diet. This would mean overall there would be an N of 8 for each diet. I am not sure this appears to have been the case though as there was no mention of repeating the trial 3 times. My guess is that the 8 athletes were just split into 3 random groups, which would leave a very low N for each group (between 2 and 3). Could the authors please explain how many athletes were in each group? Each figure should have the number of data points in each group and if possible show the individual data points (instead of just bar graphs).

I’m also unsure of what statistics were used. It was mentioned that two-way analysis of variance was used but this confuses me as the only variable appears to be carbohydrate intake.

Another concern is that there appears to be no information on the technical reproducibility of the 13C-MRS measurements. For example, were numerous readings taken per sample and then averaged? If so, what was the technical variability?

Questions

  • I’m confused with the trial setup. It was mentioned that it was a randomized crossover trial. Does this mean that all 8 athletes had to perform this experiment 3 times separate times, in which case each athlete tried each diet? If so, how long did they have between each trial to recover and return to baseline? Or if each of the 8 athletes did not undergo each of the 3 diets, how many athletes were there in each group? It would be important to have under each figure the “N” for each group.
  • Regarding the “Habitual Intakes of Energy and Macronutrients”, if the 8 athletes did not undergo each of the 3 diets separately (i.e. 3 separate trials), then are there differences between these habitual intakes for the different groups?
  • For the 24-hour time point in Figure 2, what statistics were used? I do not see how a “two-way analysis of variance” makes sense as the only variable is carbohydrate consumption. We need to know the statistical test used and the N for each group.
  • What statistics were used for Figure 3 and what was the N for each group? Again you state in your statistics that a “two-way analysis of variance was used to assess differences between the groups” but I’m a bit confused as the only variable seems to be the amount of carbohydrates eaten.

Reviewer 2 Report

Effect of different carbohydrate intakes within 24 hours after glycogen depletion on muscle glycogen recovery in Japanese endurance athletes

Nama-Motonaga et. al., Nutrients

The authors present a study of the glycogen restoration rates following exercise in Japanese males associated with three different carbohydrate level diets.  The study is generally very well organized and the results are clearly presented.  The authors provide merit justification by comparing their study, which examines Japanese participants and characteristically Japanese meals, to previous studies examining males of other ethnicities and consuming different diets.  After careful review, this manuscript appears to contain no significant flaws which  need following minor modifications.  Only one potential error in manuscript preparation is noted below.

There is no name after the word “and” in the list of authors.  Was a name left off?

Round 2

Reviewer 1 Report

I appreciate the authors changes to manuscript, the methodology is now much clearer and my concerns regarding the experimental design have been addressed.

I'm still not quite sure exactly how the statistics were performed for Figure 2 and Figure 3. For example I would've assumed Figure 2 would use repeated measures but it's possible I'm overlooking something. If possible I'd love to have a bit more detail about the stats used to help me understand if it was performed appropriately.
